# Circulating Long Non-Coding RNAs Could Be the Potential Prognostic Biomarker for Liquid Biopsy for the Clinical Management of Oral Squamous Cell Carcinoma

**DOI:** 10.3390/cancers14225590

**Published:** 2022-11-14

**Authors:** Ruma Dey Ghosh, Sudhriti Guha Majumder

**Affiliations:** 1Molecular Biology Department, Netaji Subhas Chandra Bose Cancer Research Institute, 3081 Nayabad, Kolkata 700094, India; 2Tata Translational Cancer Research Center, 14 MAR (E-W), Newtown, Kolkata 700160, India

**Keywords:** long non-coding RNA (lncRNA), oral squamous cell carcinoma, recurrence, metastasis, cancer biomarker, prognostic biomarker, liquid biopsy

## Abstract

**Simple Summary:**

Oral squamous cell carcinoma (OSCC) ranks as one of the deadliest cancers in India. Only early detection of the disease with specific prognostic subtype, is the key to reducing OSCC related death. Long non-coding RNAs (lncRNA) are master regulator of almost all biological processes. Evidences have shown that the aberrant expression of lncRNAs are found to exceed their role in different pathophysiological conditions including OSCC. The distinct expression profile of dysregulated lncRNAs in circulation could be a potential indicator to predict the OSCC-disease prognostications. The present review demonstrates that these aberrant expression of circulating lncRNAs may become powerful information regarding the blood-based biomarkers for the early prediction of disease-prognosis in OSCC. The present review also demonstrates clinical significance, limitations and challenges of circulating lncRNAs to be a potential reliable biomarker for the development of liquid biopsy technique which will be very useful, rapid, easy for clinicians for prognosis, disease monitoring, and clinical decision making to manage the treatment modality accurately in OSCC.

**Abstract:**

Long non-coding RNA (lncRNA) have little or no coding potential. These transcripts are longer than 200 nucleotides. Since lncRNAs are master regulators of almost all biological processes, recent evidence proves that aberrantly expressed lncRNAs are pathogenic for oral squamous cell carcinoma (OSCC) and other diseases. LncRNAs influence chromatin modifications, transcriptional modifications, post-transcriptional modifications, genomic imprinting, cell proliferation, invasion, metastasis, and apoptosis. Consequently, they have an impact on the disease transformation, progression, and morbidity in OSCC. Therefore, circulating lncRNAs could be the potential cancer biomarker for the better clinical management (diagnosis, prognosis, and monitoring) of OSCC to provide advanced treatment strategies and clinical decisions. In this review, we report and discuss the recent understandings and perceptions of dysregulated lncRNAs with a focus on their clinical significance in OSCC-disease monitoring and treatment. Evidence clearly indicates that a specific lncRNA expression signature could act as an indicator for the early prediction of diagnosis and prognosis for the initiation, progression, recurrence, metastasis and other clinical prognostic-factors (overall survival, disease-free survival, etc.) in OSCC. The present review demonstrates the current knowledge that all potential lncRNA expression signatures are molecular biomarkers for the early prediction of prognosis in OSCC. Finally, the review provides information about the clinical significance, challenges and limitations of the clinical usage of circulating lncRNAs in a liquid biopsy method in early, pre-symptomatic, sub-clinical, accurate OSCC prognostication. More studies on lncRNA are required to unveil the biology of the inherent mechanisms involved in the process of the development of differential prognostic outcomes in OSCC.

## 1. Introduction

Oral squamous cell carcinoma (OSCC) is recognised as one of the most common cancers, worldwide. The Surveillance, Epidemiology, and End Results (SEER) programme in the United States revealed that 54,000 new cases were diagnosed, with a death of 11,230 oral cavity cancer patients, in 2020 [1]. Recently, published data in GLOBOCAN clearly stated that globally more than 377,713 new oral cavity cancer cases and approximately 177,757 deaths have been reported in 2020 [2]. OSCC develops from the mucosal epithelial cells of the oral cavity [3]. The genetics of OSCC are complex and highly heterogeneous [4]. Surgery is the first line of treatment in OSCC. In spite of recent advanced multimodality treatment strategies (combining surgery, radiotherapy, and chemotherapy), the cure rate is around 60%, and 30–50% cases generally recur loco-regionally or distantly [4,5]. The occurrence of recurrence and metastasis are the major reasons of OSCC-induced mortality. In most cases, the recurrence is detected only after the disease has progressed to an inoperable stage, and then palliative care remains the only option [6]. The extent of the disease and the chances of treatment failure at the time of the primary tumour detection (at diagnosis) are very important in determining further prognosis [5]. Therefore, it is very important to discover novel methods using novel biomarkers to determine the disease-risk, accurately. Until now, there has been no existing molecular marker to classify OSCC subgroups, accurately.

In humans, the first-discovered lncRNA was the H19 lncRNA (derived from the IGF2 locus), and then the XIST (X-inactive specific transcript) was discovered lncRNA along with its functionality, long before the completion of the human genome project [7,8]. Recent revolutionary studies (FANTOM and ENCODE) have also demonstrated that 80% of the human genome possesses dynamic transcriptional marks and approximately 2% of these are translated into coding transcripts, and rest of the transcripts (98%) are non-coding RNAs (ncRNA) [9,10]. Therefore, ncRNAs are considered as the main key player/regulator for all normal biological functions and for any altered pathophysiological conditions which are still being investigated. MicroRNA is one of the most studied small ncRNAs. Previously we reviewed and studied the clinical implications of dysregulated miRNAs in the pathogenesis of OSCC [11,12,13]. LncRNA is another class of regulatory ncRNA transcript. It is more than 200 nucleotides to 2 Kb in length. The protein coding potential of lncRNAs is either nil or is considered to be less than 100 amino acids [14,15,16]. According to LNCipedia, a public database for lncRNA sequence and annotation, to date, approximately 127,802 annotated lncRNAs transcripts and 56,946 genes have been discovered in the human genome [17,18]. Their expressions are highly restricted to the specific tissue cell type [18]. LncRNA plays important regulatory roles in almost all physiological and pathological processes and it is thus considered as a new hallmark of cancers [19]. Cell-free lncRNAs have also been reported in different body fluids (e.g., peripheral blood plasma/serum) [20,21]. In the past decades, multiple genome-wide association studies (GWASs) have revealed that single nucleotide polymorphisms (SNPs) in lncRNA genes are predisposed and associated with the risk of carcinogenesis processes, and they may alter the disease phenotype and treatment responsiveness and patient outcome [22,23].

Earlier studies have revealed that an increased number of lncRNA are found to be dysregulated in head and neck cancer. In OSCC, abnormalities of different lncRNAs have been seen by several investigators, such as MALAT1, HOTTAIR, HOTTIP, and TUG1 [24,25,26,27,28,29,30]. LncRNA, with its tumour-promoting oncogenic drivers and its tumour suppressive role, may cause the functional dysregulation that is associated with all pathophysiological conditions in OSCC. In the present review, we provide the up-to-date information regarding the biogenesis, mechanism of actions, biological functions, overall control/regulation (gene/protein expression) and dysregulation of lncRNAs with their clinical consequences in OSCC. In this review, we elaborate and reviewed the advancements and challenges of the clinical usage of dysregulated lncRNA, either from tumour tissues or body fluids (circulating/cell free), as a molecular biomarker for the early detection and prediction of OSCC prognosis at the pre-symptomatic, sub-clinical disease stage, before it recurs. Further, in this review, we specially emphasise the clinical importance of circulating lncRNAs as a biomarker for the development of a non-invasive, nonprotein, liquid biopsy-based method for prognostication and for the novel therapeutic approach for the clinical management of OSCC.

## 2. Biogenesis of Long Non-Coding RNA

The biogenesis of lncRNA is multidimensional and dependent upon the genomic localisation of the lncRNA. So far, it is not clearly understood, therefore we need to uncover so many things for a deep insight of this process, along with the function of lncRNAs. In eukaryotes, lncRNAs are transcribed by RNA polymerase II that is typically from a poorly conserved region in the genome-like, intronic regions, and the exonic, intergenic and protein-coding regions [31]. Accordingly, lncRNAs are classified into five categories: sense lncRNA, antisense lncRNA, intronic lncRNA, intergenic lncRNA, bidirectional lncRNA, promoter lncRNA and enhancer lncRNA. Sometimes, lncRNAs share similar or overlapping biogenesis processes with protein-coding mRNAs. However, the biogenesis process of lncRNA is unique and highly specific to the order of their expression pattern. In most of cases, after being transcribed, pre-mature lncRNAs undergo unique and specific processing events, like gaining a special methyl-guanosine capping at the 5′end, a polyadenylation at the 3′end, undergoing cleavage and alternate splicing, the circularisation of RNA, the formation of small nucleolar RNA, or the formation of paraspeckles (a dynamic sub-nuclear structure) [32].

An epigenetic modification has a strong impact on the biogenesis of lncRNA. Recently, the epigenomic profiling of oesophageal squamous cell carcinoma (ESCC) revealed that the lncRNAs, like CCAT1 and LINC01503 expression, are regulated by the TP63mediated activation of its super-enhancer and promoter [33,34]. LncRNAs such as HOTTIP, FIRRE, and XIST are involved in the Histone H3 lysine 4 (H3K4) methylation-mediated transcriptional gene activation and also in the organisation of the 3D nuclear structural architecture [35]. Transcription factors binding to lncRNA leads to the formation of a nascent transcript which regulates mRNA processing by performing alternative splicing. The alternate splicing in the process of lncRNA maturation is one of the unique features that is very essential for protein diversity [32]. Some specific splicing factors facilitate the splicing process through interactions with lncRNA to form an RNA-RNA duplex with the pre-mRNA molecules. Subsequently, the splicing process of the target mRNA (gene) is achieved by altering the chromatin remodelling processes [36]. The binding of lncRNA to mRNA may enhance or delay the rate of translation or mRNA deterioration. The decoys of lncRNAs, namely, Alu transcripts (lncRNA-DNA triplex), have the ability to inhibit the transcriptional regulation [37]. Studies of transcriptomic RNA sequencing suggest that lncRNA can encode functional small RNAs [16,38]. In the cytoplasm, lncRNA are generally associated with small peptides through the ribosomes [39]. LncRNAs can be transcribed from the pseudogenes or the promoter or the intergenic regions [40]. Any alterations in the lncRNA biogenesis process lead to an alteration of its expression and its functions and their related target genes and its protein expression, which ultimately leads to a distinct pathological condition such as cancer.

## 3. Mechanism of Action

In last few decades, revolutionary changes in our understanding of genome regulation have emerged. The fundamental role of lncRNA is the synchronisation and regulation of gene/protein expression, thereby resulting in the fine-tuning of each and every physiological processes. Rarely, they contain short open reading frames (ORFs) [16]. The unique feature of lncRNAs is their ability to interact with other RNAs (e.g., mRNA, circRNA, miRNA, and others), DNA, and protein molecules, and consequently, they regulate many different biological processes at different levels in different ways. The formation of secondary RNA structures allows the lncRNAs to play the key role in controlling the adjacent (*cis*) and very distant (tans) domains in a specific chromatin region (loci) [23,41]. The functions of lncRNAs are very complex, multidimensional, and multifaceted (Figure 1). Still, they remain unclear, and current research fails to explain the sensitivity and specificity that is needed to achieve these lncRNA-mediated interactions, and the regulation of gene expressions are found to be highly cell tissue-specific in normal patients or in any patients with disease conditions like cancer.

### 3.1. LncRNA as Chromatin Regulators

LncRNAs perform their function mostly through different chromatin-based mechanisms such as signals, decoys, guides, and scaffolds in chromatin remodelling. LncRNAs play a major role in genomic imprinting. For the expression of protein coding genes, lncRNA can regulate their neighbouring (*cis*) or distant (trans) genomic environments by acting as an enhancer or a diffuser. LncRNA mediates epigenetic modifications by positioning chromatin-remodelling complexes to specific chromatin loci. It was estimated that 38% of lncRNA is found in various tissues binds to PRC2 and to other chromatin-modifying proteins like CoREST (REST corepressor 1 protein encoded by RCOR1 gene) and SMCX (also called JmjC-domain protein encoded by X-linked mental retardation gene SMCX or JARID1C gene) [19]. LncRNAs (ANRIL, XIST, KCNQ1OT1, and HOTAIR) bind to trithorax chromatin-activating complexes (TrxG) by recruiting different epigenetic modifiers to its assigned loci for chromatin remodelling [42,43].

ANRIL/CDKN2B, the antisense lncRNA in the INK4 locus, acts as a scaffold and transcriptionally silences the INK4b-ARF-INK4a locus. It binds to polycomb repressive complex 1 and 2 (PRC1 and PRC2) [44,45]. When PRC2 binds to the gene locus, it causes the spreading of the methylation marks, which is distinct for the transcriptional silencing of genes. A 17 Kb gene called X-inactive specific transcript (XIST), that is positioned on the human X chromosome, is responsible for the X chromosome dosage compensation. *Cis*-X chromosome regulation begins after the X chromosome is covered and PRC2 is recruited to its specific sites, and thus, this results in the emergence of histone H3 lysine K27 trimethylation (H3K27me3) and also causes X-linked inactivation [8,46]. LncRNA HOTAIR (HOX transcript antisense RNA) acts as a scaffold by cleaving itself to PRC2 and mediating the homeobox D cluster (HOXD) locus repression by spreading the H3K27me3 marks, and thus, this causes gene silencing [47]. HOTAIR forms many double stem-loop structures which bind to the lysine-specific demethylase1 (LSD1) and the PRC histone modification complexes [48]. KCNQ1OT1 (KCNQ1-overlapping transcript 1) antisense lncRNA, which belongs to a potassium voltage-gated channel subfamily, remains upregulated in colon cancer [49] and acts as a signal lncRNA by associating itself to G9a histone methyl-transferase and also to PRC2 [50]. When PRC2 and G9a methyl transferase are recruited to KCNQ1OT1, they mediate gene-silencing associated marks such as the demethylation of lysine 9 (H3K9me2) and lysine 27 on histone 3 [51]. KCNQ1OT1 boosts the transcriptional silencing of genes through chromatin remodelling.

### 3.2. LncRNA in Transcriptional Regulation

LncRNAs have the potential control in transcriptional regulation through modulating the expression and functions of different transcription factors, which in turn regulates different gene expression. LncRNAs can act as a co-factor of transcription factors and enzymes that are related by chromatin modification. They can regulate gene expression in *cis* (neighbouring) or in trans (distant) environments. Evf2 lncRNA recruits the transcriptional activator, DLX1, to the key DNA enhancer to repress the gene expression [42,52]. More recently, a detailed study on an ultra-conserved enhancer (UCE) uncovered that it has the lncRNA-dependent topological and transcriptional control, through complex effects, on the chromosome topology by interacting with multi-megabase distant genes. Evf2 lncRNA with Dlx5/6 forms a cloud-forming structure of UCE, which concurrently accomplishes the activation (Umad1, 1.6Mb distant) and repression of (Akr1b8, 27Mb distant) chr6 target genes, locally [42,52]. Recently, a new class of lncRNA, the eRNA (enhancer RNA), has been discovered at the gene enhancer region and is implicated mainly in transcriptional regulation [53].

LncRNA causes interaction with RNA-binding factors, namely heterogeneous nuclear ribonucleoproteins (hnRNPs). The hnRNPs then form ribonucleoproteins (RNPs) which then can act as enhancers to promote transcriptional processing by recruiting key transcription machinery proteins to their specific target gene promoters. RNPs can also cause the repression of gene transcription by attaching themselves to existing gene repressors. Fas and Blk are pro-apoptotic genes, for which, lncRNA mediates their repression by acting as a decoy for the transcriptional factor, NFYA [54].

LncRNAs play a major role in the modification of RNA polymerase (RNAP) II activity by interacting with the initiation complex, and guiding it to choose the precise promoter. It has been seen that in humans, the transcription of ncRNAs from the upstream region of the dihydrofolate reductase (DHFR) locus leads to the formation of a triplex in the promoter region, thus leading to the inhibition of the binding of the transcription factor, TFIID31 [55]. The basic components of RNAP II-dependent transcription machinery interactions with the lncRNA are transcribed by RNAP III. For further regulation, this lncRNA extracts their expression from an RNAP II-dependent transcription reaction. For example, the transcription of Alu elements bind to RNAP II in response to heat shock, eliminating the requirement of the pre-initiation complex and this causes repression by their domain interaction [56].

### 3.3. LncRNA in Post-Transcriptional Regulation

LncRNAs are potentially able to recognise their complimentary sequences, thus allowing for interactions that are responsible for the regulation of the post-transcriptional processing of mRNA. The processes in which lncRNAs are involved include capping, splicing, editing, transport, translation, and stability at various control sites. For example, the interaction of MALAT1 with splicing factors interrupts the process of alternate splicing. RNCR2 (also called MIAT or Gomafu) is an lncRNA that affects the mRNA splicing to provide a neuron-specific expression by interacting with the splicing factor 1 (SF1) and blocking the formation of spliceosome [50,56,57]. Natural antisense (NAT) lncRNAs recruit repressor complexes like PRC2 to the target genes and prompt the formation of RNA duplexes, inhibit *cis*-regulatory elements, and lead to the alternate splicing of paired genes. NAT of ZEB2 binds to the 5′splice site of an intron in 5′ UTR of ZEB2 mRNA. This intron comprises the internal ribosome entry site (IRES), which is the essential component of the translational machinery. NAT overexpression prevents splicing and increases ZEB2 expression and consequently, it down regulates the E-cadherin expression.

### 3.4. Role of lncRNA in Genomic Imprinting

Genomic imprinting is a normal epigenetic process of gene regulation by which a subset of genes can be expressed in a parent-of-origin-specific manner from one of the parental chromosomes [56]. Specific genomic loci, known as imprinting control regions (ICRs), control the genomic imprinting. Methylated and unmethylated DNA genomic imprinting regions are dependent on their parental origin for the specific expression of lncRNA genes which leads to the activation or suppression of neighbouring genes in *cis*-regulating machinery. Instead of PRC2, DNA methyltransferase plays a major role in lncRNA-mediated histone modification and DNA methylation in uniparental gene expression [58]. LncRNA and protein-coding genes are associated with imprinted clusters and are inversely expressed. AIRN and KCNQ1OT1 are examples of such lncRNAs that are responsible for the genomic imprinting of paternally inherited genes [59]. KCNQ1OT1 takes a crucial step in the long-range bidirectional repression of chromatin structures of different protein-coding genes by associating with the chromatin-modifying complexes, EED and G9A/EHMT2, and with the RNA itself. KCNQ1OT1/LIT1 is considered as an imprinted control region 2 (ICR2), which consists of at least eight genes that are expressed from the maternal allele [60]. KCNQ1OT1 silences the KCNQ1 imprinting control region by functioning like an organiser and by interacting with chromatin-modifying complexes, EED and G9A/EHMT2, and with the RNA itself [60,61,62,63,64]. Insulin-like growth factor-2 (Igf2) and insulin-like growth factor-2 receptor (Igf2r) are examples of maternally and paternally imprinted genes [65]. H19, an lncRNA plays an important role in regulating maternal imprinting which is essential for the regulation of cellular differentiation during embryogenesis in humans [66]. H19, after associating with methyl-CpG-binding-domain protein1 (MBD1), recruits histone-lysine-methyl transferase-containing complexes which form repressive H3K9 methylation marks on the targeted imprinting loci. The absence of H19-mediated maternal imprinting may cause Beckwith-Wiedemann Syndrome (BWS) and correlates with an increased risk of developing a Wilms tumour of the kidney [67,68,69,70]. The dysregulation of imprinting genes is reported in some pathological conditions, including cancer [71]. In Figure 1, we have tried to summarise the different regulatory roles of lncRNA on gene and protein expressions (Figure 1). Collectively, the above information has revealed that the functions of lncRNAs are unique, due to their ability to establish molecular interactions with proteins and all types of nucleic acids to modulate/regulate their accessibility, localisations, and functions. The multidimensional and multifaceted functional versatility and flexibility of lncRNAs has emerged in recent days [72,73]. The complete understanding of the functional plasticity of lncRNAs will give fundamental insight into the mechanisms that lncRNAs employ for gene/protein regulation and the pathophysiological processes of cancers, including OSCC.

## 4. Functional Dysregulation of lncRNAs in Oral Squamous Cell Carcinoma

An lncRNA is a new kind of gene (RNA transcript) that is without any coding proteins, but instead, has a load of complex functional regulatory access for other genes and proteins. The functional dysregulation of lncRNAs may be due to several factors, for example, the above-mentioned lncRNA-mediated regulatory mechanisms, and others [74,75,76]. The specific altered expression (functional) signature of lncRNAs can create a mark for a specific disease condition and thereby, it may serve as an independent predictor for a specific patient’s outcome with an OSCC diagnosis [41,75,76]. There are complex regulatory inter-relationships between different lncRNAs at the different stages (initiation, progression, and maintenance) of the carcinogenesis processes in different cancers. LncRNAs may be tumour suppressing or tumour promoting in nature and may ultimately lead to either the activation of tumour-promoting machineries (RNA/DNA/genes/proteins) or the deactivation of tumour-suppressor machineries (RNA/DNA/genes/proteins). Therefore, because of these altered regulations (one or more), a normal cell drives to transform into a cancer cell, and ultimately it dictates the distinct disease progression, metastasis, stemness, and treatment responsiveness in each OSCC patient. The modern cancer research has convincingly evidenced and offered a promising role for lncRNAs in the clinical management of cancers for the patients’ benefit. Previous studies have revealed that the expression of lincRNA PCAT-1 (prostate cancer associated transcript 1) is highly cancer tissue specific, whereas the expression level of circulating lncRNA PCA3 (prostate cancer antigen 3) is clinically significant in body fluids (i.e., blood) and it is well accepted for viable clinical applications as liquid biopsy biomarker for prostate cancer by the FDA (US Food and Drug Administration) [77,78]. However, lncRNA applications remain in their initial stage to be used as a predictive biomarker and currently, they are not used in clinical practices for molecular diagnostics, prognostics, and therapeutics for the clinical management of OSCC.

### 4.1. Dysregulated lncRNAs as Predictive Biomarker for OSCC Disease Management

Recent, advanced studies that have used high-throughput technologies have accelerated the rapid discovery of differentially expressed lncRNAs in tumour tissues and in other body fluids, even in OSCC and other cancers [75,78,79]. Further, recent reports have also demonstrated that the stability and relative abundance of lncRNAs are significantly high when they are in circulation (plasma/serum/saliva), therefore, these are well suited and quite accepted for the development of non-invasive liquid biopsy biomarker for OSCC prognostications [80,81]. In OSCC, a significant number of lncRNAs were found to be either upregulated or downregulated, when they were analysed for the differential expression profiling of lncRNAs in tumour tissues and in body fluids (blood, serum, plasma, and saliva) of patients when compared to the respective control (Table 1), and those lncRNAs were further evaluated for their clinical implications to be a reliable biomarker for the diagnosis, prognosis, and therapeutics of OSCC (Figure 2). Here, in this section, we give a special emphasis only to those published reports on aberrantly expressed lncRNAs in OSCC and the studies which have a significant correlation with the disease progression and outcome in patients with OSCC which were found through different search engines (PubMed, Google, different lncRNA-databases, and others) using related keywords (Table 1). Here, we have reported all these studies where a genome-wide or a targeted (single/more) profiling of lncRNAs has been conducted so far using either tumour tissues, or blood (whole blood/plasma/serum), or saliva from the patients with OSCC to find out their clinical relevance (Table 1). The data of frequently found mutations (simple somatic mutations) on the lncRNA genes of primary tumours were collected from The Cancer Genome Atlas (TCGA) database of the head and neck squamous cell carcinoma (HNSC) patient cohort (Table 2).

### 4.2. Clinical Correlations of Dysregulated LncRNAs in Primary Tumours

In conventional methods that are used to treat OSCC, the prognostic information, risk stratification, and clinical decisions are mainly derived from the histopathology report of surgical samples (from primary tumours or punch biopsy tissues) and with other imaging methods and tools that include different imaging technologies like MMR, tomographic scanning, PET-CT, X-ray, USG, etc. [4,5]. The tumour staging, lymphovascular invasion (LVI), perineural invasion (PNI), extranodal extension (ENE), depth of invasion, pattern of invasion, grade of differentiation, anatomic location, other demographic data that are associated with specific disease conditions are important indicators for decision making by a multidisciplinary team to provide the most effective clinical interventions [4,5]. The genetics of OSCC, along with intra/inter tumoural heterogeneity, are very complex and play a major role in precision oncology and differential patient outcome. So far, there is no molecular biomarker in the clinical practice for the treatment decision and disease monitoring of OSCC. The presence of some proteins markers can be detected in tumour tissues by means of immunohistochemistry (IHC). Therefore, there is an urgent need of the development of a sensitive and specific molecular biomarker to facilitate the advanced, tailor-made treatment options for disease monitoring and to attain a better clinical outcome in OSCC.

The several GWASs and other advanced studies in the last decades have identified ample number of lncRNAs, which have distinct expression signature profiles for distinct disease states in different stages of the disease progression, leading to a distinct disease outcome in OSCC. Increasing evidence discloses the emerging impact of dysregulated lncRNA expression and functional dysregulation in the pathogenic development of OSCC (Table 1 and Table 2). An earlier study has also investigated the correlation of the altered level of lncRNA expression with the risk factors and the clinicopathological factors of OSCC. The results revealed that a high level of LncRNAs AC007271.3 expression was significantly correlated with smoking history, pathological differentiation, nodal metastasis, and advanced TNM staging in OSCC. Recently, lncRNA LOLA1 was discovered to have a drastic role for promoting the transformation of malignant formations through oral leucoplakia to OSCC, and maintaining tumour progression migration and EMT via the AKT/GSK-3b pathway in OSCC [143]. A comparative study of patient tumour samples vs. normal oral mucosal tissue samples has shown that H19 lncRNA is significantly overexpressed due hypomethylation in its promoter region in tumour tissue and this is correlated with tumour grade and lower disease-free survival (DFS) in patients with OSCC [114]. One of most studied lncRNAs, HOTAIR, remains highly expressed in OSCC tissues than it is in normal healthy tissues. Upregulated HOTAIR is correlated with tumour size, TNM staging, and also with a poor prognosis of OSCC [25], and also could be a target for therapy [75,174]. The high expression of HOTTIP is seen in OSCC patients (tongue) with a T3/T4 grade tumour, distant metastasis, or in patients in clinical stages III-IV vs. a low expression in patients with T1/T2 grade tumours and with no distant metastasis or in patients in clinical stages I-II [29]. Another tumour-promoting lncRNA, LINC01929, accelerates the tumour progression by targeting the miR-137-3p/FOXC1 axis in OSCC, suggesting a novel target for OSCC therapy [137]. According to the studies of Zhou et al., the level of MALAT1 remains high in OSCC patients and it is correlated with a poor prognosis [24,30]. In a separate study, it has been shown that the overexpression of MALAT1 promotes cell proliferation and invasion by regulating the miR 101/EZH2 axis in OSCC [27]. The high level of CCAT1 lncRNA expression has also been found in OSCC and it causes a poor therapeutic outcome through sponging the activity of miR155-5p and let7b-5p [94]. LncRNA FTH1P3 remains highly expressed in OSCC and it is involved in promoting a proliferative capacity and in enhancing colony-formation [109]. FTH1P3, thus, can be a good biomarker as well as a therapeutic target for OSCC patients. The elevated LINC00673 levels and its genetic variants are associated with the development of large tumours in patients with OSCC. Authors have demonstrated that smokers are more susceptible to the risk of lymphatic spread, whereas, LINC00673 and rs9914618 single-nucleotide-polymorphism (SNP) are associated with tumour progression in the case of betel nut chewing or smoking in OSCC [130]. The higher expression of LINC00673 represents a positive correlation with tumour size, higher TNM staging, and also with recurrence, irrespective of risk factors in OSCC [131]. The upregulation of HIFCAR is seen in OSCC and it is correlated with advanced tumour staging and so, it acts as a prognostic biomarker. Fang et al. showed in their experiment that UCA1 lncRNA is responsible for increased cell proliferation and the development of cisplatin resistance and thereby, it enhances the migration of OSCC cells [175,176]. The high expression of LINC01133 was associated with less metastasis and a good prognosis [136]. Further, the tumour promoting role of TUG1 was accelerated by the sponging of miR-219 through elevated levels of FMNL2 [28]. Interestingly, in a recent study, it has been found that HOXB-AS3 has encoded a micropeptide, which is oncogenic and promotes cell proliferation and tumour progression through the activation of other oncogenes (e.g., c-Myc) in OSCC [16]. In OSCC, ANRIL is correlated with proliferation and progression. The low expression of ANRIL causes a reduction in proliferation, invasion, and migration. Thus, lncRNA ANRIL can be considered as a promising biomarker and also as a therapeutic target for OSCC patients [87].

The expression level of NKILA lncRNA remains low in OSCC and it is correlated to tumour size, clinical staging, and metastasis. So, NKILA is considered as a tumour suppressor lncRNA for disease progression and metastasis and it acts through the NF-KappaB signalling pathways in OSCC [149,150]. The expression level of lncRNA C5orf66-AS1 remains lower in OSCC tissues than it does in normal tissues [177,178]. EFGR-AS1 and its genetic variants play a significant role in treatment responses in OSCC [90]. In OSCC patients, a lower level of SOX21-AS1 was correlated with a poor prognosis [165]. A lower expression level of MEG3 is seen in OSCC tumour tissues than is seen in non-malignant tissues, and it is associated with OSCC progression and a high OSCC mortality rate [179]. Moreover, it has been found that the overexpression of MEG3 reduces the self-renewal and invasive features of cancer cells in OSCC [144]. The downregulation of PTENp1 (lncRNA pseudogene) is inversely corelated with tumour histological differentiation and progression in OSCC [155]. In addition to the critical role of lncRNA HANGA1-mediated regulation by targeting SLC2A1 in cancer cell metabolism (mitochondrial function), it added value to these RNA transcripts as a novel mechanism in the designing of therapeutic modalities [180].

Even after accumulating and analysing all this information, the research remains in a very preliminary stage and there is a need to overcome so many challenges to establish novel lncRNAs expression signatures for the early detection of OSCC diagnostics and prognostics. There are some disparities in the study designs, sample size, end-point objectives, and use of proper control, leading to variations and inconsistencies and inconclusive results. However, despite these limitations, all these studies provide the foundation for setting up the proper standard operating protocols (SOPs) to establish the lncRNA expression signature as a useful, reliable biomarker for the clinical management of OSCC, with logical interpretations for the execution of future studies using larger patient cohort.

### 4.3. Clinical Correlations of Dysregulated Circulating LncRNAs in Body Fluids

The molecular diagnostic power of circulating lncRNAs is widely accepted, worldwide due to their significantly high abundance and stable presence in all most all biological fluids including saliva and peripheral blood. The stability of circulating, cell-free lncRNAs comes from the formation of different secondary structures in combination with ribonucleoproteins and lipids, high density lipoproteins (HDLP), protective exosomes, micro-vesicles, etc., to escape the nuclease (RNase) -mediated degradation while they are in circulation [11,78]. Several published studies in the literature have already established the reliability, feasibility, and acceptability of the aberrant expression of circulating lncRNAs as a clinical biomarker for molecular diagnostics and the prognostics of other cancers [79]. In OSCC, currently, there are no circulating lncRNAs in clinical practice, but there is a significantly high sensitivity and specificity to demonstrate its proof of concept in a clinical trial for disease management.

#### 4.3.1. LncRNA Biomarker in Saliva

Saliva is one of the important, non-invasive sampling sources in OSCC diagnostics. In this regard, less invasive oral swab/brush-biopsy/scalpel/scraper biopsy methods are also very important sampling sources that are used to identify lncRNA biomarkers for OSCC diagnostics and prognostics [181,182]. Several investigators have shown that an array of ncRNAs were identified in saliva samples, and they suggested their clinical implications in OSCC [79,117,183,184]. Previously, Tang et al. demonstrated that HOTAIR and MALAT1 were significantly detected in saliva samples in OSCC patients [117]. Further, salivary HOTAIR expression was significantly upregulated in nodal metastasis when compared to the node negative-OSCC patients, suggesting that it is a rapid, non-invasive diagnostic biomarker for liquid biopsy method for OSCC management [117]. In this regard, a major drawback is the low sample size. An earlier study by Gibb et al. has shown that lncRNA NEAT-1 is overexpressed in normal mucosa when it is compared to that of OSCC mucosa [148]. Recently, Shieh et al. have studied the potential role of circulating lncRNA XIST expression in saliva and have shown that the absence of lncRNA XIST expression is associated with an increased risk of OSCC morbidity [173].

#### 4.3.2. LncRNA Biomarker in Plasma

Peripheral blood (plasma/serum) is another sampling source for a liquid biopsy for OSCC. Dong et al. revealed that the expression levels of lncRNA CASC2 remained low in the plasma samples of patients with local recurrence, but in patients without recurrence, the levels were high [93]. The overexpression of CASC2 is associated with an increased cell proliferation and thus, it can act in the prognosis of OSCC [93]. Zhang et al. have shown that lncRNA PAPAS expressions were significantly upregulated in the plasma of patients with OSCC and it can distinguish the stage 1 OSCC patients from the healthy controls [154]. Further analysis also suggested that high plasma levels of PAPAS were followed by a low overall survival rate in OSCC [154]. LncRNA NCK1-AS1 levels were also upregulated in the plasma of OSCC patients and therefore, the authors showed that it can distinguish oral ulcers from the early stages of OSCC, and claimed its early diagnostic prognostic value [146]. Further, in a study with 41 patients who underwent radical chemoradiotherapy, lncRNA HOTAIR, lincRNA-p21, and GAS5 expression were measured in their plasma samples and the results demonstrated that GAS5 showed significant association with the treatment response, whereas HOTAIR lincRNA-p21 levels in plasma did not show any conclusive results in head and neck cancers [111,112]. GAS5 expression was found to be upregulated in the plasma samples of patients with a progressive disease (poor prognosis) when it was compared to those of good clinical responsive patients [110]. Very recently, Jia et al. have identified four lncRNAs, ENST00000412740, NR_131012, ENST00000588803, and NR_038323, in plasma through microarray experiments that can be used as biomarkers for the early diagnosis and staging of OSCC. Further, they validated the expression levels of these four lncRNAs in the plasma of a larger patient cohort, with respect to the early stage (TNM I/II, 28 patients), and the advanced stage (TNM III/IV, 36 patients), and the pre-symptomatic stage (dysplasia/healthy control, 16 cases) in OSCC. Finally, a receiver operating characteristic (ROC) curves and logistic regression analysis revealed the diagnostic effects of the combined lncRNAs, and suggested that these four lncRNAs could be promising biomarkers for the early diagnosis and staging of OSCC for the benefit of clinical decision-making [101]. Earlier, HOXA11-AS, LINC00964, and MALAT1 were also identified as having potential roles as circulating liquid biopsy biomarkers in the plasma of OSCC patients, for early detection of OSCC [134].

#### 4.3.3. LncRNA Biomarker in Serum

Previously, a case-control study with 80 OSCC and 70 control individuals using serum samples demonstrated that lncRNA AC007271.3 in combination with a tumour specific growth factor (TSGF) and a squamous cell carcinoma antigen (SCCA) could be a potential circulating molecular biomarker for OSCC diagnostics. In this study, they have shown that the tumour molecular profile is actually reciprocated in the serum level, with the presence of AC007271.3 providing strong evidence to suggest that OSCC-specific lncRNAs could be released into circulation [82,83]. Recent studies have also demonstrated that lncRNA LOC284454 could be a potential serum biomarker for the diagnosis, prognosis, and therapeutic target for the clinical management of OSCC [142].

#### 4.3.4. LncRNA Biomarker in Extra-Cellular Vesicles (EVs)

Extra-cellular vesicles or exosome vesicles (EVs) from plasma, serum, or saliva could be another important source for a liquid biopsy in OSCC. Recent studies have shown that circulating lncRNAs are well protected from endosomal RNase degradation while they are in circulation [183]. So far, according to our knowledge, the lncRNA expression profile in EVs that were collected from saliva/plasma/serum have not been studied in OSCC, however, there are ample amounts of similar information for other cancers [76].

Recent studies have shown that the source of these spectrum of circulating lncRNAs have been found due to the direct reciprocal secretion from the tumour cells, apoptotic cells, necrotic cells and/or due to the consequences of an immunogenic reaction upon the pathophysiological or metabolic response to cancer. Although, cancer cells are known to evade apoptosis through a variety of mechanisms, it is already established that these tumour-associated circulating lncRNAs are released in the extra-cellular space by tumour cells along with protective lipids and proteins, RNA-binding proteins, neucleophosmin (NPM1), Argonaute (AGO) proteins, high density lipoproteins, or sometimes without any binding partner within the exosome vesicles to avoid nuclease activity when they are in circulation [11]. Several investigators have predicted specific circulation lncRNAs–cancer associations [78,185]. The expression of circulating lncRNAs sometimes may be different from the expression profile in OSCC tumour tissue. Overall, circulating lncRNAs have been shown to constitute a promising biomarker for use in a liquid biopsy, which is a more rapid, minimally invasive, robust, cost effective, easy, reliable, and consistent method for both OSCC diagnosis and prognosis (Table 3). Further, it is quite obvious to suggest that circulating lncRNAs along with other interacting partners may constitute future diagnostic tools or therapeutic targets with a higher sensitivity and specificity for better OSCC disease management.

## 5. Clinical Impact of Dysregulated lncRNAs in OSCC Prognosis

### 5.1. LncRNA in Lymph Node Metastasis and Distant Metastasis

There is evidence that invasiveness and metastasis, as well as EMT (epithelial-mesenchymal transition) and MET (mesenchymal-epithelial transition), are controlled by lncRNAs. Cancer cells, as a result of intercellular communication, acquire new properties of plasticity, stem-like features and thus, they become prone to therapy resistance [186,187]. The loss of E-cadherin expression is a decisive indicator that ensures consecutive transformation, EMT, and metastatic invasion. Lymph node metastasis (LNM) is considered as an essential prognostic factor, whereas the distant metastasis is considered as a very common condition and it is related to the advanced stages in OSCC patients [4,5]. LncRNAs are correlated with this nodal metastasis as well as distant metastasis (Table 1).

According to tissue analysis studies, MALAT1 shows a positive correlation with regional LNM in OSCC patients. Further reports from recent studies showed that MALAT1 plays an important role in regulating the metastatic (LNM/distant) ability by inducing the EMT, invasion, and migration of cancer cells in OSCC [24,27,30]. However, it has been found that a high expression of HOTTIP lncRNA in OSCC is correlated to distant metastasis that is caused by the disease [29]. Several studies showed that a high level of HOTAIR lncRNA expression is significantly associated with an induced EMT and frequently with the occurrence of LNM, invasion, and migration in OSCC [26,75,174]. A research study by Fang et al. proved that UCA1 lncRNA is associated to LNM and thus, this results in an increased migration of OSCC cells [175,176]. It has been shown that if the expression of ANRIL, LOLA1, and LINC00673 lncRNAs is increased in OSCC, then the patient outcome is generally associated with a poor prognostication and metastasis in OSCC [87,130,143]. Another study has revealed that there is a positive correlation between TUG1 and LNM and it can inhibit the invasion of OSCC [168]. The overexpression of LINC00662 in OSCC tissues is associated with tumour size, LNM, and TNM staging [128]. H19 lncRNA mediates its metastatic signalling cascade through the sponging of miR-148a-3p and it causes the release of DNA methyltransferase enzyme (DNMT1) by inducing EMT and lowering the E-cadherin expression in OSCC [114]. In OSCC, FOXCUT lncRNA is associated with the gene, FOXC1, which regulates EMT and expressions of MMPs and VEGF-A genes, and thereby, also regulates cell proliferation and migration and metastasis [106]. FTH1P3 (Ferritin heavy chain 1 pseudogene 3) lncRNA of the ferritin heavy chain gene family is also related to the progression and metastasis of OSCC cells [109]. In OSCC, the downregulation of NKILA lncRNA causes the elimination of the inhibitory effect on NF-κβ, promotes EMT, and so it gets involved in migration and invasion [149]. ROR lncRNA is responsible for cellular migration, invasion, and metastasis in OSCC [139]. On the other hand, LINC01133 is found to be a positive marker for no metastasis and a good prognosis in OSCC [136].

### 5.2. LncRNAs in Disease Free Survival (DFS) and Overall Survival (OS)

Clinical treatments have made progress but the overall survival rate, as per the studies, is still about 50–60% in OSCC. The expression of lncRNA is important for providing prognostic information which will thereby help to predict the specific disease outcome, like DFS and OS (Table 1). Lower MEG3 expression enhances tumour progression and it is correlated with a high mortality rate and a poor overall survival, while MEG3 overexpression induces apoptosis [75,188]. It is statistically proven that lower MEG3 expression is associated with a shorter OS than those with higher expression. Overexpressed FOXCUT lncRNA is associated with a poor overall survival in OSCC patients [106]. LncRNA PANCR is positioned adjacent to the PITX2 gene and it is seen that PANCR remains highly methylated in OSCC [152]. The hypermethylation of PANCR leads to an increase in the death rate of patients. The high methylation of PITX2 in OSCC is correlated with p16 expression and a higher survival rate. Other reports have revealed that the overexpression of FTH1P3 lncRNA is associated with progression and metastasis and therefore, it contributes to a poor OS and is correlated with a high mortality rate and a poor overall survival [30]. In TSCC patients, LINC00673, and HOTTIP lncRNAs remain highly expressed and cause induced migration and invasion and they are negatively associated with OS and disease-free survival [29,30].

## 6. LncRNA as a Therapeutic Target in OSCC

LncRNA could be a promising important therapeutic target for OSCC disease management. Although the detail functional regulatory controlling mechanism of lncRNAs are yet to understand properly, several clinically important lncRNAs came into clinical trial for diagnostic and prognostic purpose for human cancers. In this regard, ANRASSF1, PCA3 (NCT01024959), HULC (DRKS00017517), CCAT1 (NCT04269746), H19 (NCT04767750), ANRIL, MALAT1 are currently in clinical trial (trial number given in brackets) for cancer patients. These are important potential candidate lncRNAs, either came into clinical practice or yet to come [77,189]. PCA3 lncRNA (Progensa) is already in clinical practice for diagnosis of prostate cancer. CCAT1 and PCA3 and HULC are using as non-invasive or minimally-invasive diagnostic biomarker using body fluids (urine/blood) for detection of cancer.

In OSCC, researchers have started to focus on how lncRNA can be useful as a target for therapies. Advanced studies suggested that in OSCC, lncRNAs, like HOTAIR, HOTTIP, UCA1, H19, LOLA1, TUG1, FOXCUT, FTH1P3, LINC00673 and PTENP1 can be potential therapeutic targets to manage the better patient outcome in OSCC. On the other hand, tumour suppressor like MEG3, LINC01133 and NKILA lncRNAs could be the potential positive indicator for therapeutic target for better OSCC patient outcome. In OSCC progression, lncRNA act as tumour suppressor as well as oncogenes. So, if the expression levels of lncRNA are altered accordingly then they can be used as good therapeutic targets. As lncRNA could act as good prognostic biomarker it could be useful in developing new treatments for OSCC patients.

## 7. Challenges and Future Perspective of Circulating lncRNAs as Prognostic Biomarker for Liquid Biopsy in OSCC

The above-mentioned research has revealed numerous lncRNAs were aberrantly expressed (upregulated/downregulated), that were either found in tumours tissue or in circulating body fluids (saliva/plasma/serum) in OSCC (Table 1 and Table 2). Although, the results were not very consistent among the different independent studies, the systemic validation of some potential candidate lncRNAs is required through a multicentric randomised control trial using a larger patient cohort with well-characterised patients with OSCC. The specificity and sensitivity should be very high to be a good clinical biomarker. A well-characterised reference control and a reliable internal control should be present for the normalisation of the results. The detailed mechanisms of lncRNAs and their complex regulatory network in a biological system are not fully understood. A huge number of dysregulated lncRNA expression signatures (upregulated/downregulated) were observed in recent cancer research, and these could provide a larger window for the development of specific lncRNA-based biomarkers (Table 1 and Table 2) because of their greater abundance when they are compared to protein-coding mRNAs for the early detection of OSCC diagnosis, prognosis, and therapeutic targets. The lncRNA is highly stable in tissues and body fluids and its expression is highly tissue specific. Further, there are involvements of lncRNAs in different multidimensional, diverse cellular signalling and regulation in gene expression during different stages of the carcinogenesis process. Therefore, lncRNAs could be the best candidate for the development of clinical biomarkers for the early detection at the pre-symptomatic, sub-clinical disease stage for its better management with advanced treatment options, which might be useful for tailor-made precision oncology in OSCC. However, there are still so many challenges and limitations to this research, and so, rigorous validation and evaluations are required for their clinical applications for OSCC.

The current knowledge has established that circulating lncRNAs in saliva or in blood (plasma/serum) may constitute a potential biomarker for the early detection of OSCC for diagnosis and could also be valuable biomarker for disease prognosis in OSCC. Several studies of other cancers have already established that some circulating lncRNAs serve as potential biomarkers to predict disease evolution and eventual clinical outcome [77,78,185]. In OSCC-based cancer research, circulating lncRNAs still remain in their initial phase, and there is much more to be explored before their useful clinical applications. Further, in this regard, there are some points to be noted: (i) the use of whole blood is usually not recommended, whereas the use of plasma and serum would be fine for the accurate quantification of circulating lncRNAs. This is because if the patient is experiencing inflammation or metabolically active, lncRNA expression profiles in blood cells (red and white) may interfere with their variable results [190,191]. (ii) An equal volume of starting material (saliva/plasma/serum) from different patients, may not produce the same RNA quantity and quality. Therefore, it would be better to start with large sample volume to get the good amount of yield of RNA, with better quality to be gained upon the extraction procedure [185]. (iii) The quality, quantity, and integrity of RNA that is extracted from body fluids could be an issue for using high-throughput techniques. Therefore, we advocate for the use of a proper reference gene or an endogenous control for data normalisation [185,192]. (iv) The inclusion of exogenous spike-in control and usage of specialised instruments like a Qubit, a concentrator, or sensitive RQ-PCR machines are recommended to obtain good quality data from the biological fluids [30,192]. Although there are many challenges and limitations, it is very clear that circulating lncRNAs have been shown to constitute innovative therapeutic targets and reliable sensitive and specific biomarkers for the development of a liquid biopsy technique for the early prediction of OSCC disease prognostication.

## 8. Conclusions

Recent, large-scale comprehensive studies on lncRNA biology with clinical samples have revealed that the transition of lncRNA-based diagnostics to lncRNA-based therapeutics has already started to develop for the management of human diseases. In this regard, the main challenge and the important issue is that the selection of these initial lncRNA selections remain unbiased during the processing and interpretation of the high-throughput data. Still, an extensive validation is required through a wet-lab experimental validation, and this is very important part to conduct before we proceed further.

In this review, we documented the clinically importance of lncRNAs in OSCC. LncRNA acts as a regulative factor for different carcinogenesis processes (Table 1). LncRNA functions in multiple ways at different levels (Figure 1). As per the research revelations, it is clear that the variations in expression levels of different lncRNAs may have a distinct correlation with the differential OSCC progression and metastasis. Therefore, it is clear that lncRNAs have the potentiality to be used for the screening, diagnosis, prognosis, and risk assessment of OSCC, and to decide its treatment strategy and the mode of disease monitoring to predict the patient’s outcome, with respect to factors like recurrence, metastasis, disease-free-survival, and overall survival in patients with OSCC. LncRNAs like MALAT1, HOTAIR, HOTTIP, MEG3, CASC2, and FTH1P3 also act as prognosticators in OSCC. Many other lncRNA like MALAT1, TUG1, and AFAP1-AS1 are also seen to be associated with nodal metastasis in OSCC. LncRNA also gives evidence for its relationship with overall survival and disease-free survival. LncRNA still remains a challenging topic for research and new consolidations are essential for OSCC. Overall, as described in the present review, circulating lncRNAs could be promising, reliable, robust, cost-effective biomarker for use in a liquid biopsy for disease monitoring, prognostication to check residual disease, and treatment responsiveness with an easy sampling method before and after treatment (surgery/chemotherapy/radiotherapy) in OSCC.

## Figures and Tables

**Figure 1 cancers-14-05590-f001:**
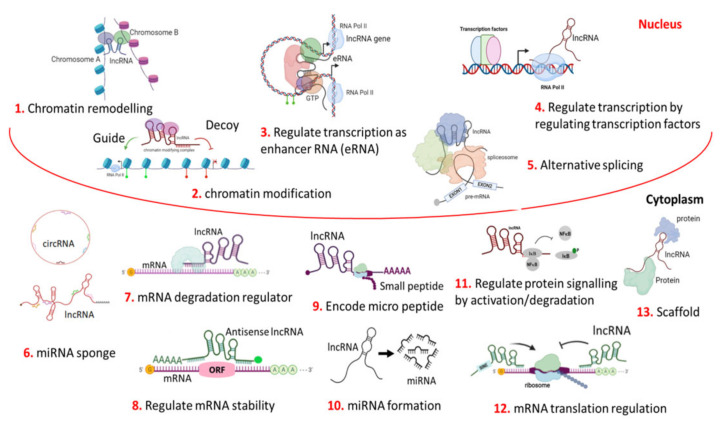
Different, complex, multi-dimensional functions of LncRNA. LncRNA mainly functions as signals, decoys, scaffolds, and as guides. Thus, lncRNA can act as a regulator for chromatin remodelling their environment (neighbouring or distant through intra or inter-chromosomal interaction) by positioning chromatin modifiers at the chromosomal level in the genome architecture (1). LncRNA can regulate gene expression by recruiting a chromatin-modifying complex either to activate or repress the neighbouring genes at the transcription level (2). It can also regulate transcription as an enhancer RNA or eRNA (3), or by the binding and/or activating of transcription factors to the promoter region (4). It can regulate pre-mRNA processing through the alternative splicing of mRNAs (5). In cytoplasm, at a circular form or linear form, it can sponge (silence) the function of miRNA (6). It can also act as an mRNA degradation regulator (7); mRNA stability can also be regulated by lncRNA (8). It can code for small micro peptide (9). They can generate miRNA by degrading themselves (10). It can regulate signalling cascades in different physiological pathways through the activation/deactivation of proteins in cancer cells (11). It can regulate the mRNA translation (12). Architectural scaffolding is another important function of lncRNAs. The LncRNA-mediated scaffolding of protein (RNA-protein/ribonucleoprotein) structures are called paraspeckles, which are found in interchromatin space. This is also found in several shared pathways in the cytoplasm (13).

**Figure 2 cancers-14-05590-f002:**
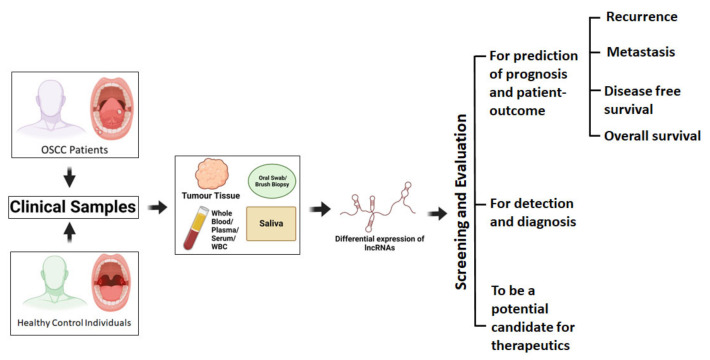
The schematic presentation of the entire process by which screening, evaluation, and clinical significance of altered lncRNA expression signatures have been conducted and analysed for the detection of the disease and for the early prediction of prognosis (recurrence, metastasis, treatment responsiveness, survival, etc.) and for the development of novel, efficient therapeutics in OSCC using clinical samples collected from patients (tumour, blood, saliva, etc.) and healthy control individuals (blood, saliva, etc.).

**Table 1 cancers-14-05590-t001:** Dysregulated lncRNAs in OSCC.

LncRNA ID	Expression Status	Sample Source	Target Gene	Clinical Significance	Reference
**AC007271.3**	up	Tumour, Serum	NFκB, miR-125b-2-3p, Slug	cell proliferation, EMT, Metastasis;	[82,83]
**AC132217.4**	up	Tumour	KLF8, IGF2	metastasis, migration, invasion;	[84]
**ADAMTS9-AS2**	down	Tumour	miR-600, Ezh2, AKT signaling	marker for early diagnosis and metastasis;	[85]
**AFAP1-AS1**	up	Tumour	miR-145, HOXA1	cell proliferation, progression;	[86]
**ANRIL**	up	Tumour	TGFβ, Smad, Bcl-2	proliferation and progression, tumour progression;	[87]
**BANCR**	up	Tumour	MAPK signalling	cell proliferation, progression, migration;	[88]
**BLACAT1**	up	Tumour	PSEN1	cell proliferation, progression, radioresistance;	[89]
**C5orf66-AS1**	down	Tumour		diagnostic marker, treatment response;	[90]
**CASC15**	up	Tumour	miR-124, miR-33a-5p	proliferation and invasion, progression, and metastasis;	[91]
**CASC2**	down	Tumour, Plasma	miR-21, miR-31-5p, KANK1	local recurrence, cisplatin responsiveness;	[92,93]
**CCAT1**	up	Tumour	miR-181a, Wnt/β-catenin, DDR2, ERK, AKT, miR-55-5p, let7b-5p	poor therapeutic outcome;	[94]
**CCAT2**	up	Tumour	Wnt/β-catenin, Ccnd1, Myc, GSK3β	poor prognosis, metastasis, migration, invasion;	[95]
**CRNDE**	up	Tumour	miR-384	proliferation and invasion, progression and metastasis;	[96]
**DANCR**	up	Tumour	miR-216a-5p	histological grade, clinical staging, lymph node metastasis;	[97]
**DLEU1**	up	Tumour	miR-149-5p, CDK6, HA, CD44 signalling	disease progression, diagnostic marker;	[98]
**DNM3OS**	up	Tumour	miR-204-5p, HIP1	disease progression, migration;	[99]
**ELDR**	up	Tumour	ILF3, cyclinE1 signalling	disease progression, cell proliferation;	[100]
**ENST00000412740**	up	Plasma		biomarker for early diagnosis and staging of OSCC;	[101]
**ENST00000527317**	down	Tumour		poor median PFS and OS;	[102]
**ENST00000583044**	down	Tumour		poor median PFS and OS;	[102]
**ENST00000588803**	up	Plasma		biomarker for early diagnosis and staging of OSCC;	[101]
**FER1L4**	up	Tumour	miR133a, Prx1	disease progression;	[103]
**FGD5-AS1**	up	Tumour	MCL1, miR-153-3p	proliferation and migration, invasion;	[104]
**FLJ22447**	up	Tumour	IL-33	disease progression;	[105]
**FOXCUT**	up	Tumour	FOXC1	cell proliferation, migration, metastasis, poor survival;	[106]
**FOXD2-AS1**	up	Tumour	E2F-G2-M checkpoint	migration, pathological grade, poor disease prognosis;	[107,108]
**FTH1P3**	up	Tumour	miR-224-5p, Fizzled5	progression, metastasis, high mortality rate, poor overall survival;	[109]
**GAS5**	down	Tumour, Blood, Plasma	miR-21, PTEN, FoxO1, miR-1297, GSK3β	cell proliferation, migration, EMT, metastasis, treatment responsiveness;	[110,111,112]
**H19**	up	Tumour	miR-let-7	tumorigenesis, metastasis, poor prognosis, low disease-free-survival, prognostic biomarker;	[113,114]
**HAS2-AS1**	up	Tumour	TGFα, HIF-1α, Nfκb	poor prognosis, invasion, EMT	[115]
**HNF1A-AS1**	up	Tumour	STAT3, Notch signalling	poor prognosis, EMT, migration	[116]
**HOTAIR**	up	Tumour, Saliva, Plasma	Ezh2, E-cadherin	TNM staging, poor prognosis, marker for early detection;	[25,26,111,112,117]
**HOTTIP**	up	Tumour	miR-124-3p, HMGA2, Wnt/β-catenin	migration, invasion, distant metastasis, poor overall survival, poor disease-free-survival;	[29]
**HOXA10**	up	Tumour		risk factor, tumour grade;	[118]
**HOXA11-AS**	up	Tumour, Plasma	miR-98-5p, YBX2	disease progression;	[119]
**HOXB-AS3**	up	Tumour		cell proliferation and tumour progression;	[16]
**HOXC13-AS**	up	Tumour	miR-378g, HOXC13 axis	cell proliferation, migration, EMT;	[120]
**JPX**	up	Tumour	miR-944, CDH2 axis	cell proliferation, migration, invasion;	[121]
**LEF1-AS1**	up	Tumour	LATS1, Hippo signalling	migration, metastasis;	[122]
**LINC00152**	up	Tumour	miR-139-5p	poor prognosis, metastasis, migration, invasion, EMT;	[123,124]
**LINC00284**	up	Tumour	miR-211-3p, MAFG axis, FUS, KAZN axis	cell proliferation, migration;	[125]
**LINC00460**	up	Tumour	Peroxiredoxin-1, miR-4443	poor prognosis, metastasis, migration, invasion, EMT;	[126,127]
**LINC00662**	up	Tumour		tumour size, LNM, and TNM staging;	[128]
**LINC00668**	up	Tumour	miR-297, VEGFA signaling	poor prognosis;	[129]
**LINC00673**	up	Tumour		betel nut association, TNM staging, recurrence, migration, invasion, poor overall survival, poor disease-free-survival;	[130,131]
**LINC00941**	up	Tumour	CAPRIN2, Wnt/β-catenin	disease progression;	[132]
**LINC00958**	up	Tumour	miR-211-5p, CENPK axis, JAK, STAT3 signaling	shorter overall survival;	[133]
**LINC00964**	up	Plasma		marker for early detection;	[134]
**LINC01116**	up	Tumour	miR-136, FN1	disease progression, invasion, and migration;	[135]
**LINC01133**	up	Tumour	GDF15	less metastasis, good prognosis;	[136]
**LINC01929**	up	Tumour	miR-137-3p, FOXC1 axis	tumour progression;	[137]
**LINC02487**	down	Tumour	USP17, SNAI1 axis	migration, invasion, cancer metastasis;	[138]
**Linc-ROR**	up	Tumour	Oct4, Nanog, Sox4, Klf4, cMyc	cellular migration, invasion, and metastasis;	[139]
**Lnc-p23154**	up	Tumour	Glut1	poor prognosis, metastasis, migration, invasion, EMT;	[140]
**lnc-WRN-10:1**	down	Tumour		poor median PFS and OS;	[102]
**LOC100506114**	up	CAFs	RUNX2, GDF10 signaling	proliferation and migration;	[141]
**LOC284454**	up	Serum		early diagnostic marker;	[142]
**LOLA1**	up	Tumour	AKT, GSK3β pathway	tumour progression, migration, invasion, EMT;	[143]
**MALAT1**	up	Tumour, Plasma, Saliva	Cks1, Wnt/β-catenin, miR-101, Ezh2 axis	EMT, marker for early detection and poor prognosis, metastasis;	[24,27,30,117]
**MEG3**	down	Tumour	miR-421, Dnmt3B	high mortality rate and poor overall survival, tumour recurrence, metastasis. tumour suppressor;	[144]
**MIR31HG**	up	Tumour	HIF-1α	cellular migration, invasion, and metastasis;	[145]
**NCK1-AS1**	up	Plasma	miR-100	marker for early detection and metastasis;	[146]
**NEAT1**	up	Tumour, Saliva		therapeutic target, marker for early detection;	[147,148]
**NKILA**	down	Tumour	NFκB signalling	tumour volume, weight, proliferation, invasion, migration, metastasis;	[149,150]
**NR_038323**	up	Plasma		biomarker for early diagnosis and staging of OSCC;	[101]
**NR_104048**	down	Tumour		poor median PFS and OS;	[102]
**NR_131012**	up	Plasma		biomarker for early diagnosis and staging of OSCC;	[101]
**ORAOV1-B**	up	Tumour	NFκB, TNF-α signalling	lymph node metastasis, invasion, migration, metastasis;	[151]
**PANCR**	up	Tumour		Hypermethylation and poor survival;	[152]
**PANDAR**	up	Tumour		Metastasis, migration, invasion, poor prognosis;	[153]
**PAPAS**	up	Plasma	TGFβ1	Biomarker for diagnosis, poor overall survival;	[154]
**PTENp1**	down	Tumour	miR-21, PTEN	histological differentiation and progression;	[155]
**PVT1**	up	Tumour	miR-150-5p, GLUT1	poor prognosis;	[156]
**RC3H2**	up	Tumour	miR-101-3p, Ezh2	metastasis, migration, invasion;	[157]
**SLC16A1-AS1**	up	Tumour	CCND1	histological grade, overall survival;	[158]
**SNHG12**	up	Tumour	miR-326, E2F1	proliferation and migration, invasion, EMT;	[159]
**SNHG16**	up	Tumour	CCND1	cell proliferation, viability, migration and EMT;	[160]
**SNHG17**	up	Tumour	miR-384, ELF1, CTNNB1, Wnt/β-catenin, miR-375, PAX6 axis	disease progression;	[161,162]
**SNHG20**	up	Tumour	miR-197, LIN28 axis	oncogenesis and tumourigenesis;	[163]
**SNHG3**	up	Tumour	Wnt/β-catenin, NFYC	proliferation, migration;	[164]
**SOX21-AS1**	down	Tumour		poor prognosis;	[165]
**TIRY**	up	Tumour	miR-14, Wnt/β-catenin	proliferation, migration;	[166]
**TTN-AS1**	up	Tumour	miR-411-3p, NFAT5	disease progression;	[167]
**TUG1**	up	Tumour	miR-219, FMNL2, Wnt/β-catenin, cyclin D1, cMyc	tumour promoting, lymph node metastasis;	[28,168]
**UCA1**	up	Tumour	miR-138-5p, CCR7, Wnt/β-catenin	proliferation, migration, invasion, glycolysis metabolism;	[169,170]
**VENTXP1**	down	Tumour	miR-205-5p, ANKRD2, NFκB	poor survival;	[171]
**XIST**	up	Tumour, Saliva	miR-27b-3p	cell proliferation, cisplatin resistance.	[172,173]

**Table 2 cancers-14-05590-t002:** Simple somatic mutations found in lncRNA-genes in OSCC. Data collected from TCGA-HNSC patient cohort from TCGA database.

Name	Symbol	Simple Somatic Mutation Affected (%)	Number of Mutations	Mutation Details (Changes in DNA)
X inactive specific transcript	XIST	7/37 (18.92%)	7	chrX:g.73841444C>T chrX:g.73844493T>C chrX:g.73845294A>G chrX:g.73844293G>C chrX:g.73843798G>T chrX:g.73842529A>G chrX:g.73844433G>C
AL109984.1	AL109984.1	3/37 (8.11%)	3	chr20:g.52088576G>A chr20:g.52084589C>T chr20:g.52085000C>G
family with sequence similarity E5	FAM27E5	3/37 (8.11%)	3	chr17:g.22299660G>C chr17:g.22298919G>A chr17:g.22298929C>A
chromosome 8 open reading frame 31	C8orf31	3/37 (8.11%)	3	chr8:g.143043244G>A chr8:g.143043089T>C chr8:g.143043028C>G
RNF217 antisense RNA 1 (head to head)	RNF217-AS1	2/37 (5.41%)	2	chr6:g.124910865G>A chr6:g.124910984G>C
spermatogenesis associated 8	SPATA8	2/37 (5.41%)	2	chr15:g.96783707G>A chr15:g.96784195C>T
long intergenic non-protein coding RNA 482	LINC00482	2/37 (5.41%)	2	chr17:g.81304684C>A chr17:g.81305023G>A
AC092718.9	AC092718.9	2/37 (5.41%)	3	chr16:g.81149700C>G chr16:g.81147473C>T chr16:g.81147475T>A
long intergenic non-protein coding RNA 1588	LINC01588	2/37 (5.41%)	2	chr14:g.50005654C>T chr14:g.49992302T>C
SUGT1P4-STRA6LP readthrough	SUGT1P4-STRA6LP	1/37 (2.70%)	1	chr9:g.97294104G>A
chromosome 14 putative open reading frame 177	C14orf177	1/37 (2.70%)	1	chr14:g.98716342delAG
long intergenic non-protein coding RNA 1559	LINC01559	1/37 (2.70%)	1	chr12:g.13376270C>T
chromosome 8 open reading frame 86	C8orf86	1/37 (2.70%)	1	chr8:g.38528390C>T
MIR1-1HG antisense RNA 1	MIR1-1HG-AS1	1/37 (2.70%)	1	chr20:g.62546276C>T
HLA complex group 27	HCG27	1/37 (2.70%)	1	chr6:g.31202764C>T
long intergenic non-protein coding RNA 1600	LINC01600	1/37 (2.70%)	1	chr6:g.2623640G>A
long intergenic non-protein coding RNA 1098	LINC01098	1/37 (2.70%)	1	chr4:g.177975927delTT
long intergenic non-protein coding RNA 173	LINC00173	1/37 (2.70%)	1	chr12:g.116534694C>T
mir-99a-let-7c cluster host gene	MIR99AHG	1/37 (2.70%)	1	chr21:g.16391630G>A
chromosome 9 putative open reading frame 106	C9orf106	1/37 (2.70%)	1	chr9:g.129322444delCCAGTTCT…
long intergenic non-protein coding RNA 2877	LINC02877	1/37 (2.70%)	1	chr3:g.153484635C>G
chromosome 20 putative open reading frame 197	C20orf197	1/37 (2.70%)	1	chr20:g.60070819G>A
long intergenic non-protein coding RNA 2870	LINC02870	1/37 (2.70%)	1	chr10:g.132448053G>A
long intergenic non-protein coding RNA 1565	LINC01565	1/37 (2.70%)	1	chr3:g.128573545delC
myelodysplastic syndrome 2 translocation associated	MDS2	1/37 (2.70%)	1	chr1:g.23627122C>A
chromosome 11 putative open reading frame 40	C11orf40	1/37 (2.70%)	1	chr11:g.4573322G>A
family with sequence similarity 153 member C	FAM153CP	1/37 (2.70%)	1	chr5:g.178055978G>A
BACH1-IT1	BACH1-IT1	1/37 (2.70%)	1	chr21:g.29351669C>G

**Table 3 cancers-14-05590-t003:** Clinically important potential candidate circulating lncRNAs for liquid biopsy in OSCC.

Biotype	Circulating lncRNAs	Suggested Clinical Implications	References
PLASMA	CASC2	Prognosis, treatment response	[93]
	ENST00000412740,	Diagnosis and prognosis	[101]
	ENST00000588803,	Diagnosis and prognosis	[101]
	GAS5	Prognosis, treatment response	[112]
	HOXA11-AS	Diagnosis and prognosis	[134]
	LINC00964	Diagnosis and prognosis	[134]
	MALAT1	Diagnosis and prognosis	[134]
	NCK1-AS1	Diagnosis, prognosis	[146]
	NR_131012	Diagnosis and prognosis	[101]
	NR_038323	Diagnosis and prognosis	[101]
	PAPAS	Prognosis	[154]
SERUM	AC007271.3	Prognosis	[82,83]
	LOC284454	Diagnosis	[142]
SALIVA	HOTAIR	Prognosis	[117]
	MALAT1	Prognosis	[117]
	NEAT-1	Prognosis	[148]
	XIST	Prognosis, treatment response	[173]

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
