# Peer review of "Circulating Long Non-Coding RNAs Could Be the Potential Prognostic Biomarker for Liquid Biopsy for the Clinical Management of Oral Squamous Cell Carcinoma"

_cancers, 2022, doi:10.3390/cancers14225590_

Round 1
Reviewer 1 Report
Circulating long non-coding RNAs are discussed in the current review article by Ghosh and Majumder as a potential predictive biomarker for liquid biopsy for the clinical therapy of oral squamous cell carcinoma. The author of the current review came to the conclusion that lncRNA can serve as a potent prognostic biomarker and that it may one day serve as a therapeutic target for OSCC cancer. Overall, the review is written extremely effectively and may be accepted as is without any more significant revisions. However, there are a few recommendations and issues that might be addressed to improve the review's quality.
- Without a doubt, over the past few years, circulating long non-coding RNAs have caught the attention of researchers and clinicians who hope to use them as prognostic biomarkers in clinical management. However, the main challenge with working with circulating long non-coding RNAs is to isolate or identify them at very early stages. Since cancer cells are known for evading apoptosis, the author needs to discuss
- Before drawing any final conclusions, the majority of the review's findings or results must first be confirmed in a lab setting.
- It would be fantastic if the author could demonstrate a direct association between dysregulated lncRNAs in OSCC and LncRNA ID from the TCGA database.
- The excellence of Figure 2: It is necessary to improve the schematic representation of the procedure for evaluating dysregulated lncRNA in patient samples; the OSCC sub type section is unclear.
- The legend of f Figure 1. Function of LncRNA needs to be more explanatory.
Author Response
To
The Editor,
Cancers.
Dated: 16th August, 2022
Kolkata
Dear Editor,
I am extremely happy that our review manuscript (Manuscript ID: cancers-1835660) entitled “Circulating long non-coding RNAs could be the potential prognostic biomarker for liquid biopsy for the clinical management of oral squamous cell carcinoma.” authored by Ruma Dey Ghosh and Sudhriti Guha Majumder has received favourable reviewer’s comments.
In the revised manuscript, we have tried our level best to address almost all the concerns of the editor and reviewers and to improve our manuscript. Please find the attached document for revised version of the manuscript.
Here, I am answering to Editor’s comments point-to-point below:
Comment 1 # Please use the version of your manuscript found at the above link for your
revisions.
Answer: Thanks to the Editor and Reviewers for critical comments and suggestions and giving us the opportunity to improve our manuscript.
The specified file has been taken to revise of our manuscript.
Comment 2: (I) Please check that all references are relevant to the contents of the
manuscript.
Answer: All references have been checked thoroughly. Some are added and some are deleted in the revised version.
Comment 3: (II) Any revisions to the manuscript should be marked up using the “Track
Changes” function if you are using MS Word/LaTeX, such that any changes can
be easily viewed by the editors and reviewers.
Answer: The revised manuscript is in “track change” mode, therefore changes can be easily viewed by editors and reviewers.
Comment 4: (III) Please provide a cover letter to explain, point by point, the details
of the revisions to the manuscript and your responses to the referees’
comments.
Answer: Two rebuttal cover letters have been prepared with all explanations and answers point by point separately to the editor’s comments and reviewer’s comments.
Comment 5: (IV) If you found it impossible to address certain comments in the review
reports, please include an explanation in your rebuttal.
Answer: Thank you for your suggestion. Explained the same in our rebuttal.
Comment 6: (V) The revised version will be sent to the editors and reviewers.
Answer: Thank you for your support.
Comment 7: *(VI) Please provide us with institutional emails. If you do not have
institutional emails, please provide us with a shirt CV for both authors.
Answer: Official email address is given in the corresponding address. (rumadeyghosh.nscri@gmail.com).
Dr. Ruma Dey Ghosh:
Dr. Ruma Dey Ghosh is a Senior Scientist at Netaji Subhash Chandra Bose Cancer Research Institute (NCRI), Kolkata, India. She is a well-known cancer biologist with more than 10 years of independent cancer research experience (principal investigator). Previously, she worked as Principal Investigator position at Tata Translational Cancer Research Center, Kolkata, India. Dr. Dey Ghosh’s research focuses around the research problem which have a translational impact in various cancer disease management. Mainly to explore the biology of different non-coding RNA-mediated regulation of gene expression and its alteration during disease progression and to understand their role in differential prognostic outcome in patients with cancers.
Sudhriti Guha Majumder:
Sudhriti Guha Majumder joined Dr. Ruma’s Lab for her 6 months internship at Tata Translational Cancer Research Center, Kolkata.
Comment 8: If one of the referees has suggested that your manuscript should undergo
extensive English revisions, please address this issue during revision.
Answer: In the revised manuscript, English language correction has been done thoroughly.
Now, I am answering to reviewer’s comments point-to-point below:
Answers to the Reviewer 1’s comments:
Comment 1: Circulating long non-coding RNAs are discussed in the current review article by Ghosh and Majumder as a potential predictive biomarker for liquid biopsy for the clinical therapy of oral squamous cell carcinoma. The author of the current review came to the conclusion that lncRNA can serve as a potent prognostic biomarker and that it may one day serve as a therapeutic target for OSCC cancer. Overall, the review is written extremely effectively and may be accepted as is without any more significant revisions. However, there are a few recommendations and issues that might be addressed to improve the review's quality.
Answer: We are thankful to the reviewer for appreciation and for critical comments and suggestions and giving us the opportunity to improve our manuscript.
Comment 2: Without a doubt, over the past few years, circulating long non-coding RNAs have caught the attention of researchers and clinicians who hope to use them as prognostic biomarkers in clinical management. However, the main challenge with working with circulating long non-coding RNAs is to isolate or identify them at very early stages. Since cancer cells are known for evading apoptosis, the author needs to discuss
Answer: Thanks to the reviewer for your appreciation and favorable comments pointing out our limitations. In the revised manuscript we have reconstructed our manuscript accordingly. The issue has been discussed and incorporated in point 4.3 (Clinical correlations of dysregulated circulating LncRNAs in body fluids) and in other related areas in this review. Thanks to the reviewer for valuable suggestions.
Comment 3: Before drawing any final conclusions, the majority of the review's findings or results must first be confirmed in a lab setting.
Answer: This is not an original research article; therefore, conclusions were made mostly on the basis of published research in past decades. On the basis of these current knowledge in this field, we are developing our hypothesis through constructing a rational clinical question for future research for patient benefit. We will try to validate this by using retrospective and prospective patient cohort from Indian subcontinent.
Laboratory validation is not possible in this current manuscript, we will definitely confirm our conclusion in our future manuscript as original research article with our won experimental data. Hope, reviewer will understand our limitations in this case.
Comment 4: It would be fantastic if the author could demonstrate a direct association between dysregulated lncRNAs in OSCC and LncRNA ID from the TCGA database.
Answer: Thanks to the reviewer for valuable suggestion. From the TCGA database, here, we include the information regarding the frequently occurred simple somatic mutations in the lncRNA genes in OSCC. Accordingly, Table 2 has been constructed and included in the revised manuscript.
In the current manuscript we have voyaged through various dysregulated long non-coding RNAs (lncRNAs) reported to responsible for the pathogenesis, progression and specific outcome of OSCC. Here, we have extensively used Pubmed and Google search engine and collected relevant data by using different key words in this regard. We include all studies based on clinical samples like, tumour tissues, and other body fluids like blood plasma, serum, exosome vesicles, saliva etc. and discuss about the dysregulation of lncRNAs in OSCC. The beauty of our current study is that the curated information in this manuscript is not limited to the high throughput RNA-seq data (TCGA), we have considered all studies that reported only one or few dysregulated lncRNAs in OSCC. Here, we have also include differentially expressed (DE) dysregulated lncRNAs from genome wide lncRNA profiling through NGS-based RNA-sequencing, or through lncRNA expression-microarray platform using clinical samples of patients with OSCC. We reevaluate and correlate all the information of dysregulated lncRNAs expression and demonstrate in a meaningful manner with the clinical significance in patient diagnosis, prognosis and therapeutic targets in OSCC disease management.
Comment 5: The excellence of Figure 2: It is necessary to improve the schematic representation of the procedure for evaluating dysregulated lncRNA in patient samples; the OSCC sub type section is unclear.
Answer: Thanks to the reviewer for critical comments and suggestions and giving us the opportunity to improve our manuscript. The Figure 2 has been replaced with the new figure and clear figure legends in the revised manuscript.
Comment 6: The legend of f Figure 1. Function of LncRNA needs to be more explanatory.
Answer: In the revised manuscript we have reconstructed our manuscript accordingly to overcome our limitations. The detailed explanations of functions of lncRNAs in the Figure 1 have been included in the revised manuscript.
Answers to the Reviewer 2’s comments:
Comment 1: In the manuscript entitled “Circulating long non-coding RNAs could be the potential prognostic biomarker for liquid biopsy for the clinical management of oral squamous cell carcinoma”, the authors discussed the impacts of lncRNAs on OSCC transformation, progression and morbidity. This article is constructed to present a summary of our recent knowledge regarding the general information of lncRNAs and their correlations with OSCC progression. However, the current evidence, limitation, critiques and perception on these OSCC-associated lncRNAs expression signatures as molecular liquid biopsy biomarker for the early prediction are not touched in this work to fit the scope of their review title and the Special Sections targeted by this manuscript. Reorganization and rearrangement are required.
Answer: We are thankful to the reviewer for pointing out our flaws and weaknesses and giving us the opportunity to improve our manuscript. We apologies for our mistakes. In the revised manuscript we have reconstructed our manuscript accordingly to overcome our limitations. The current evidence, limitations, critiques, challenges, perception have been included and discussed on clinical significance of the dysregulated lncRNA expression-signature to be a biomarker for liquid biopsy method for OSCC disease management and monitoring.
English language correction has been done thoroughly.
Comment 2: As mentioned above, the context of the manuscript does not fit well to the current title, especially considering the relevant terms “liquid biopsy” and “biomarker”. More emphasis should be put on the existence and alterations of OSCC-associated lncRNAs in a variety of body fluids in recent reports, and the current evidence, limitation, critiques and perception on these OSCC-associated lncRNAs expression signatures as diagnostic and/or prognostic biomarkers.
Answer: Revised manuscript has been modified as recommended by the reviewer and more emphasis has been given to discuss the OSCC-associated lncRNAs in a variety of body fluids, and the relevance to evaluate the novel molecular biomarker for liquid biopsy method in clinical practice in OSCC treatment protocol. In the revised manuscript, throughout we have attempted to balance to fit current title and its clinical correlation with pragmatism, highlighting both the strengths and weaknesses of using lncRNA in routine clinical practice.
Section 4 has been restructured as per suggestion.
Sub-section 4.3. “Clinical correlations of dysregulated circulating LncRNAs in body fluids” with other sub-subsections has been included in the revised manuscript.
Section 7. “Challenges and future perspective of lncRNAs as prognostic biomarker in OSCC” has been included in the revised manuscript.
Accordingly, changes in the abstract, introduction, other section has been modified to fit well to the current title. The entire manuscript has been revised to make it entirely objective and comprehensive.
Comment 3: Table 1. To make it more concise and informative, some categories could be further divided and specified. For example, “Clinical impact” could be divided into molecular mechanisms, pathways and target genes, OSCC subtypes analyzed, metastatic sites, presence in body fluids, association with prognosis, and so on to make the description clear and organized. As for “DOI/Article ID”, the format coherence and consistency are required. Also, inclusion of the reference number annotated in the context will be greatly helpful for comparison.
Answer: Thanks to the reviewer for giving us the opportunity to improve our manuscript with the help of critical comments, suggestions and pointing out our weakness. Table 1 has been restructured as suggested by the reviewer. The suggested points like, Sample source, Target genes, pathways, signaling cascade, clinical significance, association with prognosis have been included in the Table 1. References are also restructured and included as suggested by the reviewer. Reference check has been done extensively throughout the manuscript including the mentioned Table 1.
Comment 4: Section 4. The structure of this section, which contained an introduction of “Dysregulation of LncRNAs in oral squamous cell carcinoma” and a sub-section “4.1 LncRNAs as predictive biomarker for OSCC disease management”, could be further rearranged. In sub-section 4.1, instead of simple descriptions about each lncRNAs, choosing categories (such as body fluids, OSCC subtypes, metastatic sites, molecular pathways and target genes, and so on) matched with Table 1 to provide a clear way to organize all the information collected.
Answer: We thanks to the reviewer for suggesting improvements. Recommendations for modifications have been accepted and implemented in the revised manuscript.
Section 4 has been restructured and reorganized as suggested
Sub-section 4.3. “Clinical correlations of dysregulated circulating LncRNAs in body fluids” with other sub-subsections has been included in the revised manuscript.
Section 7. “Challenges and future perspective of lncRNAs as prognostic biomarker in OSCC” has been included in the revised manuscript.
Accordingly, changes in the abstract, introduction, other section has been modified to fit well to the current title.
Looking forward to hearing your favourable comments,
Sincerely yours,
Ruma Dey Ghosh
Dr. Ruma Dey Ghosh, Ph.D,
Netaji Subhas Chandra Bose Cancer Research Institute,
3081, Nayabad, Kolkata -700094, India.
E-mail: deyrumai@yahoo.co.in

Reviewer 2 Report
In the manuscript entitled “Circulating long non-coding RNAs could be the potential prognostic biomarker for liquid biopsy for the clinical management of oral squamous cell carcinoma”, the authors discussed the impacts of lncRNAs on OSCC transformation, progression and morbidity. This article is constructed to present a summary of our recent knowledge regarding the general information of lncRNAs and their correlations with OSCC progression. However, the current evidence, limitation, critiques and perception on these OSCC-associated lncRNAs expression signatures as molecular liquid biopsy biomarker for the early prediction are not touched in this work to fit the scope of their review title and the Special Sections targeted by this manuscript. Reorganization and rearrangement are required.
Remarks:
1. As mentioned above, the context of the manuscript does not fit well to the current title, especially considering the relevant terms “liquid biopsy” and “biomarker”. More emphasis should be put on the existence and alterations of OSCC-associated lncRNAs in a variety of body fluids in recent reports, and the current evidence, limitation, critiques and perception on these OSCC-associated lncRNAs expression signatures as diagnostic and/or prognostic biomarkers.
2. Table 1. To make it more concise and informative, some categories could be further divided and specified. For example, “Clinical impact” could be divided into molecular mechanisms, pathways and target genes, OSCC subtypes analyzed, metastatic sites, presence in body fluids, association with prognosis, and so on to make the description clear and organized. As for “DOI/Article ID”, the format coherence and consistency are required. Also, inclusion of the reference number annotated in the context will be greatly helpful for comparison.
3. Section 4. The structure of this section, which contained an introduction of “Dysregulation of LncRNAs in oral squamous cell carcinoma” and a sub-section “4.1 LncRNAs as predictive biomarker for OSCC disease management”, could be further rearranged. In sub-section 4.1, instead of simple descriptions about each lncRNAs, choosing categories (such as body fluids, OSCC subtypes, metastatic sites, molecular pathways and target genes, and so on) matched with Table 1 to provide a clear way to organize all the information collected.
Author Response

(The authors gave the same response as above.)

Round 2
Reviewer 2 Report
The authors have reconstructed their manuscript nicely. Some of the following suggestions may help to further improve this work.
Remarks:
1. Table 1 and Section 4.3 have been restructured as suggested. Several of the relevant references in Section 4.3 could be included in Table 1 to make it more comprehensive. Otherwise, considering that the main focus of this review is the liquid biopsy biomarker, it will be helpful to organize Section 4.3 into a Table about the identification of these circulating lncRNAs in liquid biopsy application to provide a handy reference to the audience.
2. Section 4.3. A recent publication reporting salivary lncRNA XIST expression associated with OSCC seems to be relevant to this section (PMID: 34640640, Journal of Clinical Medicine) and this field.
3. Section 7. Lines 628-644. The discussion could be more informative if appropriate references could be included.
Author Response
To
The Editor,
Cancers.
Dated: 29th August, 2022
Kolkata
Dear Editor,
I am extremely happy that our review manuscript (Manuscript ID: cancers-1835660) entitled “Circulating long non-coding RNAs could be the potential prognostic biomarker for liquid biopsy for the clinical management of oral squamous cell carcinoma.” authored by Ruma Dey Ghosh and Sudhriti Guha Majumder has received favourable reviewer’s comments.
Here, I am answering to Reviewer’s comments point-to-point below:
Answers to the Reviewer 2’s comments:
Comment #: The authors have reconstructed their manuscript nicely. Some of the following suggestions may help to further improve this work.
Answer: we are grateful to the reviewer for your appreciation and critical comments with valuable suggestions. We have tried our level best to improve our revised manuscript as per reviewer suggestions.
Comment 1: Table 1 and Section 4.3 have been restructured as suggested. Several of the relevant references in Section 4.3 could be included in Table 1 to make it more comprehensive. Otherwise, considering that the main focus of this review is the liquid biopsy biomarker, it will be helpful to organize Section 4.3 into a Table about the identification of these circulating lncRNAs in liquid biopsy application to provide a handy reference to the audience.
Answer: Thanks to the reviewer for pointing out our flaws and mistakes and giving us the opportunity to improve our manuscript.
Table 1 and section 4.3 have been checked thoroughly and revised accordingly. Relevant references in section 4.3 have been included in Table 1.
Additionally, Table 3 has been created as a handy reference for take home message of section 4.3- regarding dysregulated circulating lncRNAs in different body fluids and their clinical significance in OSCC.
Thanks for your valuable suggestions.
Comment 2: Section 4.3. A recent publication reporting salivary lncRNA XIST expression associated with OSCC seems to be relevant to this section (PMID: 34640640, Journal of Clinical Medicine) and this field.
Answer: The suggested reference has been incorporated in point 4.3 (Clinical correlations of dysregulated circulating LncRNAs in body fluids) and in other related areas in this review.
Comment 3: Section 7. Lines 628-644. The discussion could be more informative if appropriate references could be included.
Answer: In this section, all relevant, appropriate references have need incorporated in the revised manuscript as suggested by the reviewer.
Please find the attached document for re-revised version of the manuscript. In this version, we have tried our level best by addressing all the concerns of the reviewer 2 to improve our manuscript.
Looking forward to hearing your favourable comments,
Sincerely yours,
Ruma Dey Ghosh
Dr. Ruma Dey Ghosh, Ph.D,
Netaji Subhas Chandra Bose Cancer Research Institute,
3081, Nayabad, Kolkata -700094, India.
E-mail: deyrumai@yahoo.co.in
